# Nitrogen and phosphorus losses via surface runoff from tea plantations in the mountainous areas of Southwest China

Xingcheng Huang[1,2], Darong Zhen[3], Xiaona Lu[4], Yarong Zhang[1,2], Yanling Liu[1,2], Yu Li[1,2]*, Taiming Jiang[2]*

1 Institute of Soil and Fertilizer, Guizhou Academy of Agricultural Sciences, Guiyang, Guizhou, China, 2 Scientific Observing and Experimental Station of Arable Land Conservation and Agricultural Environment (Guizhou), Ministry of Agriculture and Rural, Guiyang, Guizhou, China, 3 Guizhou Agricultural Ecology and Resource Protection Station, Guiyang, Guizhou, China, 4 Meitan Agricultural and Rural Bureau, Zunyi, Guizhou, China

* liyu083110@163.com (YL); jtm532@163.com (TJ)

**Data Availability Statement:** All relevant data are within the manuscript and its Supporting Information files.

## Abstract

Nowadays, there has been a rapid expansion of tea plantations in the mountainous areas of southwest China. However, little research has focused on the pollution problems caused by the losses of nitrogen and phosphorus from tea plantations in this area. Therefore, a field experiment was conducted using the runoff plots in situ monitoring method following farmers' conventional management from 2018 to 2020 in Guizhou Province, southwest China. The characteristics of nitrogen and phosphorus losses from tea plantation in the mountainous area were clarified, and the effect of rainfall intensity on the nitrogen and phosphorus losses were explored. 298 natural rainfall events with a total rainfall of 2258 mm were observed during the 2-year observation period, and erosive rainfall accounted for 78.1% of the total rainfall. The total surface runoff amount was 72 mm, and the surface runoff coefficient was 3.19%. The total nitrogen (TN) and total phosphorus (TP) concentrations in the surface runoff ranged from 0.68 to 14.86 mg·L$^{-1}$ and 0.18 to 2.34 mg·L$^{-1}$, respectively. The TN and TP losses from tea plantations were 1.47 kg N ha$^{-1}$ yr$^{-1}$ and 0.210 kg P ha$^{-1}$ yr$^{-1}$. Rainfall intensity directly and significantly affected the surface runoff and nitrogen and phosphorus loss. Where 72.6% of the cumulative rainfall, 92.5% of the total surface runoff amounts, 87.4% of total nitrogen loss, and 90.5% of total phosphorus loss were observed in rainfall events above 10 mm. Taken together, the results provide scientific guidance for quantifying the characteristics of nutrient loss in subtropical mountain tea plantations.

## Introduction

Nitrogen and phosphorus are the main pollutants of eutrophication, which have a serious impact on the aquatic ecosystem [1, 2]. China have being one of the most eutrophic regions attracting more and more attention in the world-wide-concern [3–5]. This is chiefly due to the high frequency, high intensity, and wide range of nitrogen and phosphorus emissions from

**Funding:** This study was supported by the Program of Science and Technology Plan of Guizhou Province (20201Y119 and 20201Y119) and the Program of Science and Technology Innovation Talents Team of Guizhou Province (20185604).

**Competing interests:** The authors have declared that no competing interests exist.

agricultural sources in China [6, 7]. According to the data of the Second National Survey of Pollution Sources of China, 46.5% of the total nitrogen and 67.2% of the total phosphorus in water bodies are contributed by agricultural sources [8]. However, because of the randomness of the emission time and the universality of the occurrence range of nitrogen and phosphorus pollutants from agricultural sources, the emission process is controlled by differences in topography, land use, and climate on a regional scale [9–11]. Thus, the impact of nitrogen and phosphorus losses from agricultural sources on water quality is particularly complex, which must be systematically evaluated under particular environmental and crop conditions.

Tea tree (*Camellia sinensis*) is a widely grown cash crop in the tropics and subtropics of the world. China is the world's largest tea producer, with annual harvested area accounting for 64.65% of the global total in 2021 [12]. As excessively high nitrogen and phosphorus fertilizer rates were used in tea plantations [13], nitrogen and phosphorus losses from tea plantation has been a hot spot for agricultural non-point pollution in China. Especially in southwest China, where tea trees are typically planted on sloping farmland of 6°-35° and has a typical monsoon climate, rainfall is a major dynamic source of surface runoff and nutrient loss on slopes, which may load to surface runoff and nitrogen and phosphorus discharge into water body [14]. However, the intensity and process of nitrogen and phosphorus losses in tea plantations on sloping farmland still lacks systematic understanding in these regions. Thus, to advance our understanding of sustainable tea production and water quality protection, more information on the characteristics and patterns of nitrogen and phosphorus losses from tea plantation in mountainous area of southwest China is urgently needed.

Therefore, the objectives of the present study were: (1) to evaluate the intensity and dynamic process of nitrogen and phosphorus losses via surface runoff from tea plantations ecosystems in southwest China and (2) to evaluate the effects of rainfall intensity on the surface runoff and nitrogen and phosphorus losses were also discussed. This study is of great significance for the objective assessment of characteristics of nutrient loss in subtropical mountain tea plantation.

## Materials and methods

### Study site

The study site was located in the Meitan National Agricultural Science and Technology Park, in Meitan County, Guizhou Province, China (27.71°N, 107.54°E, elevation of 883 m a.s.l). This region has a typical mountain terrain, and most of tea trees was planting on sloping fields of 6°-35°. This study site is characterized as a subtropical monsoon climate (exhibiting full sunshine, abundant precipitation, and four distinct seasons), and with an annual precipitation amount of 1112 mm and about 86% of the precipitation occurring in the rainy season from April to October. The annual average air temperature is15.2°C. The soil was classified as Orthic Acrisols [15]. The properties of surface soil (0–20-cm depth) were pH 5.10, organic carbon 21.10 g kg$^{-1}$, total nitrogen 1.53 g kg$^{-1}$, and total phosphorus 0.66 g kg$^{-1}$.

### Experimental design

A field runoff plot experiment was conducted to assess the characteristic of nitrogen and phosphorus losses via surface runoff from tea plantation in the mountainous areas of southwest China. Three runoff plots 2.6 m wide and 12 m long situated on a 12.5° slope with lengths parallel to the slope were replicated. The area of runoff plots was 31.2 m$^2$. The spacing of each plot was 6 m. The boundaries of each plot were made of bricks and cement to avoid runoff from leaving or entering the plot. At the lower end of each plot, a volumetrically calibrated tank

with a depth 1.2 m and a diameter of 1 m. The runoff water would overflow automatically to the tank.

The tea trees (*Camellia sinensis* cv. Qianmei 601.) were planted in 2008. A typical row of tea trees has canopies 0.8 m high and 0.9 m wide, with a 0.4 m interrow space between the two canopies of trees. The tea field was fertilized at common application rates of 342.5 kg N ha$^{-1}$ yr$^{-1}$ and 49.1 kg P ha$^{-1}$ yr$^{-1}$ in two additions (added with urea during the spring application and compound fertilizer during the winter application). Compound fertilizers were applied as band applications in the interrow space between canopies with widths of approximately 0.2 m and then incorporated into the soil to a depth of approximately 0.1 m. Urea was applied evenly on the soil surface on the interrow space and under the tree canopy, which is the conventional practice in tea cultivation. Other field practices, such as tea harvesting and weeding, were carried out identical to the local tea planting management regime over the experimental period.

## Sample collection and analyses

This study was performed for two consecutive years, from September 2018 to August 2020. AG1000 Weather Stations (Campbell Scientific, Inc., USA) were installed near the experimental area to record daily meteorological data (8:00 AM to 8:00 AM) during the study period. According to daily rainfall intensity, rainfall events are divided into light rain (<10 mm in a day), moderate rain (between 10 and 24.9 mm), heavy rain (between 25 and 49.9 mm) and rainstorm (≥50 mm). Discharge of the surface runoff was measured following every daily runoff event (8:00 AM to 8:00 AM) during the experimental durations, and runoff samples were collected from each plot. Depths of runoff water in tanks were measured to calculate the runoff volume first, and the ratio of the runoff volume to the plot area was calculated to obtain the runoff (mm). Then, the tanks were thoroughly stirred with clean bamboo poles, and multi-point sampling was conducted at different parts and depths. Clean carboys (500-ml HDPE plastic bottles) were used to store samples in the field. Once collected, runoff samples were stored at −4°C in a cooler and transported to the laboratory for further chemical analysis. After sampling, the runoff water from each tank was drained. The tanks were washed three times and rinsed with clean water in preparation for the next sampling.

All runoff samples were generally analyzed within one week after collection. The unfiltered runoff samples were used for total nitrogen (TN) and total phosphorus (TP) concentrations determination following the potassium persulfate oxidation method [16] using SmartChem 140 Discrete Auto Analyzer (AMS-Westco Scientific Instruments, Inc., Rome, Italy). Aliquots of the runoff samples were filtered through 0.45-μm filters for dissolved nitrogen (DN) and dissolved phosphorus (DP) determination [17]. The concentrations of particulate nitrogen (PN) and particulate phosphorus (PP) were calculated as the difference between TN and DN, and between TP from DP, respectively.

## Statistical analysis

Nitrogen and phosphorus daily loads in surface runoff were calculated for the individual plots by multiplying the nitrogen and phosphorus concentrations in each runoff sample by the runoff volume. The values were then summed to generate 2-year loads from September 2018 to August 2020. The loss rate of fertilizer nitrogen and phosphorus was calculated as TN and TP losses dividing by nitrogen fertilizer and phosphorus fertilizer input, respectively.

Data preparation and calculation were performed using Microsoft Excel 2019 (Microsoft Corp., Redmond, WA, USA). Data were tested for normality and homogeneity of variance. The figures were created using Microsoft Excel 2019 software. ANOVAs were performed with rainfall intensities and observation year as the dominant effects. The pathway analysis was

carried out to evaluate the effect of the daily rainfall intensity on surface runoff, the concentrations of nitrogen and phosphorus, and total loss amounts, using AMOS 24 (SPSS, Inc., Chicago, IL, USA). Duncan's new multiple range test was used to test the significance of difference. A $p<0.05$ was considered significant.

## Results

### Environmental variables

In this study, a total of 298 natural rainfall events were observed during the observation period, including 185 and 113 rainfall events observed in 2018–2019 and 2019–2020, respectively (Fig 1 and S1 Table). According to the rainfall intensity, 237 light rains, 39 moderate rains, 18 heavy rains, and 4 rainstorm rainfall events were observed, respectively (Fig 1 and S2 Table). Rainstorm events were observed once in 2018–2019 and three occasions in 2019–2020. The maximum rainfall event observed on May 19, 2020, with a rainfall of 109 mm. The annual rainfall for 2018–2019 was 1262 mm, which was much higher than that for 2019–2020 (996 mm), showing large interannual variabilities (S1 Table, $p<0.05$). During the monitoring period, the cumulative rainfall of light rain, moderate rain, heavy rain, and rainstorm were 618 mm, 699 mm, 647 mm, and 294 mm, respectively. The erosive rainfall events were 45 and 32 times in 2018–2019 and 2019–2020, and erosive rainfall amounts were 1001 mm and 761 mm, respectively (S1 Table). According to the rainfall intensity, 16 erosive rainfall events was observed under light rain, with erosive rainfall of 123 mm, and erosive rainfall events were observed under all moderate rain, heavy rain, and rainstorm. During the observation period, the minimum, maximum, and mean daily air temperatures were −2.5°C, 28.5°C, and 15.4°C, respectively (Fig 1).

### Surface runoff discharge

During the monitoring period, 77 surface runoff events were generated, including 45 in 2018–2019 and 32 in 2019–2020 (Fig 2 and S1 Table). The daily surface runoff was between 0.11 and 3.25 mm, and the maximum runoff occurred on May 19, 2020 (Fig 2). During the monitoring period, the total surface runoff amount was 72 mm. There were significant differences in the total amount of surface runoff between two monitoring years were noted (41 mm in 2018–2019 and 31 mm in 2019–2020, S2 Table, $p<0.05$). The daily surface runoff coefficient was between 1.73% and 10.65%, and the maximum runoff coefficient occurred on January 25, 2020. The

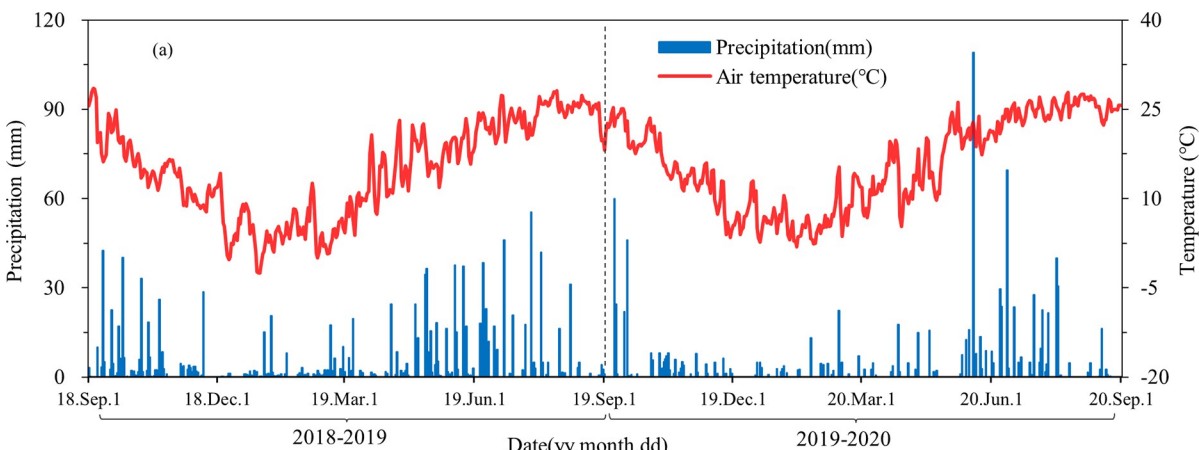

**Fig 1. Temporal changes in air temperatures and daily precipitation in the tea plantations from September 2018 to August 2020.**

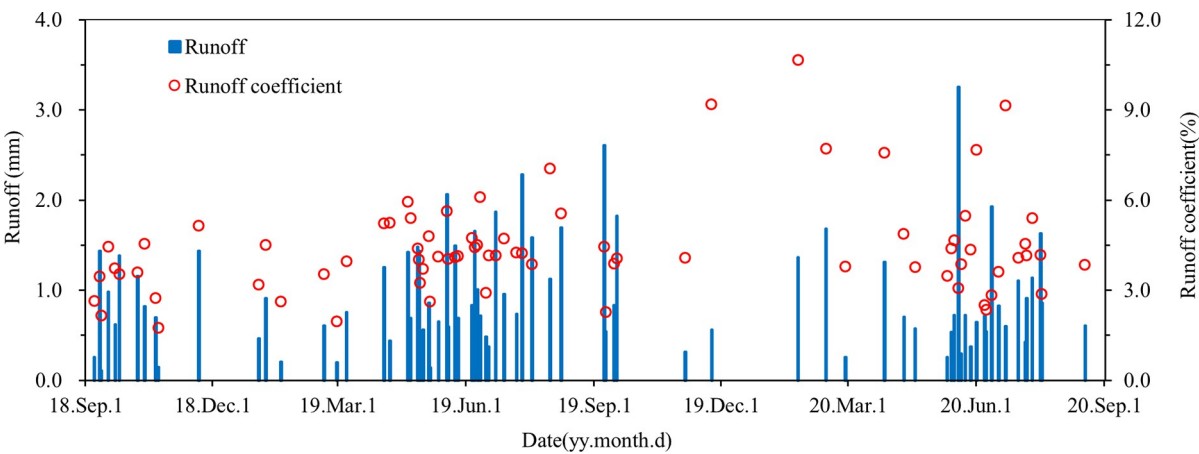

**Fig 2. Daily variations in the surface runoff and runoff coefficient in the tea plantations from September 2018 to August 2020.**

surface runoff coefficient was 3.19% during the monitoring period, with the runoff coefficient being 3.29% in 2018–2019 and 3.07% in 2019–2020 (S3 Table). According to the rainfall intensity, 16 surface runoff events were observed under light rain, with a runoff of 5 mm, and 39,18 and 4 surface runoff events were observed under moderate rain, heavy rain, and rainstorm, with a runoff of 31 mm, 26 mm, and 10 mm, respectively. The runoff coefficients were significantly different among different rainfall intensities, with light rain, moderate rain, heavy rain, and rainstorm were 0.87%, 4.46%, 3.94% and 3.42%, respectively (S4 Table, $p < 0.05$).

## Nitrogen and phosphorus concentrations in surface runoff

Overall, the TN and TP concentrations in surface runoff presented distinct fluctuations. During the monitoring period, the TN and TP concentrations in surface runoff ranged from 0.68 to 14.86 mg·L$^{-1}$ and 0.18 to 2.34 mg·L$^{-1}$, respectively (Fig 3). The results show that the TN and TP concentrations in 56 runoff waters were higher than the Class V limits specified in the environmental quality standards for surface water (TN 2.0 mg·L$^{-1}$ and TP 0.4 mg·L$^{-1}$, GB3838-2002) among the 77 surface runoff events, and the over-standard rate was 72.7%.

During the observation period, the nitrogen and phosphorus concentrations in surface runoff water were significantly different under different rainfall intensities ($p < 0.05$). The TN and TP concentrations in runoff water decreased first and then increased with increased rainfall intensities. Under rainstorm events, the TN and TP concentrations in runoff water were significantly higher than those of moderate rain, and heavy rain events (Table 1). The discharge of nitrogen and phosphorus were mainly in the form of dissolved nitrogen and dissolved phosphorus, accounting for 82.09% to 86.35% and 62.23% to 70.21% of the TN and TP concentrations under different rainfall intensities, respectively.

## Nitrogen and phosphorus losses

The daily loads of TN and TP of each surface runoff event their contributions rate to the cumulative losses were shown in Fig 4. The TN daily loads ranged from 0.72 to 490.24 g N ha$^{-1}$, and the TP daily loads ranged from 0.25 to 31.99 g P ha$^{-1}$. The highest daily loads of TN and TP were observed on May 19, 2020, with contribution rate were 16.69% and 7.61% to the cumulative losses of TN and TP, respectively. Furthermore, the loss of nitrogen and phosphorus presented seasonal changes. The contribution rate of nitrogen and phosphorus losses in the rainy season (April to October each year) were 89.10% and 83.77%, respectively.

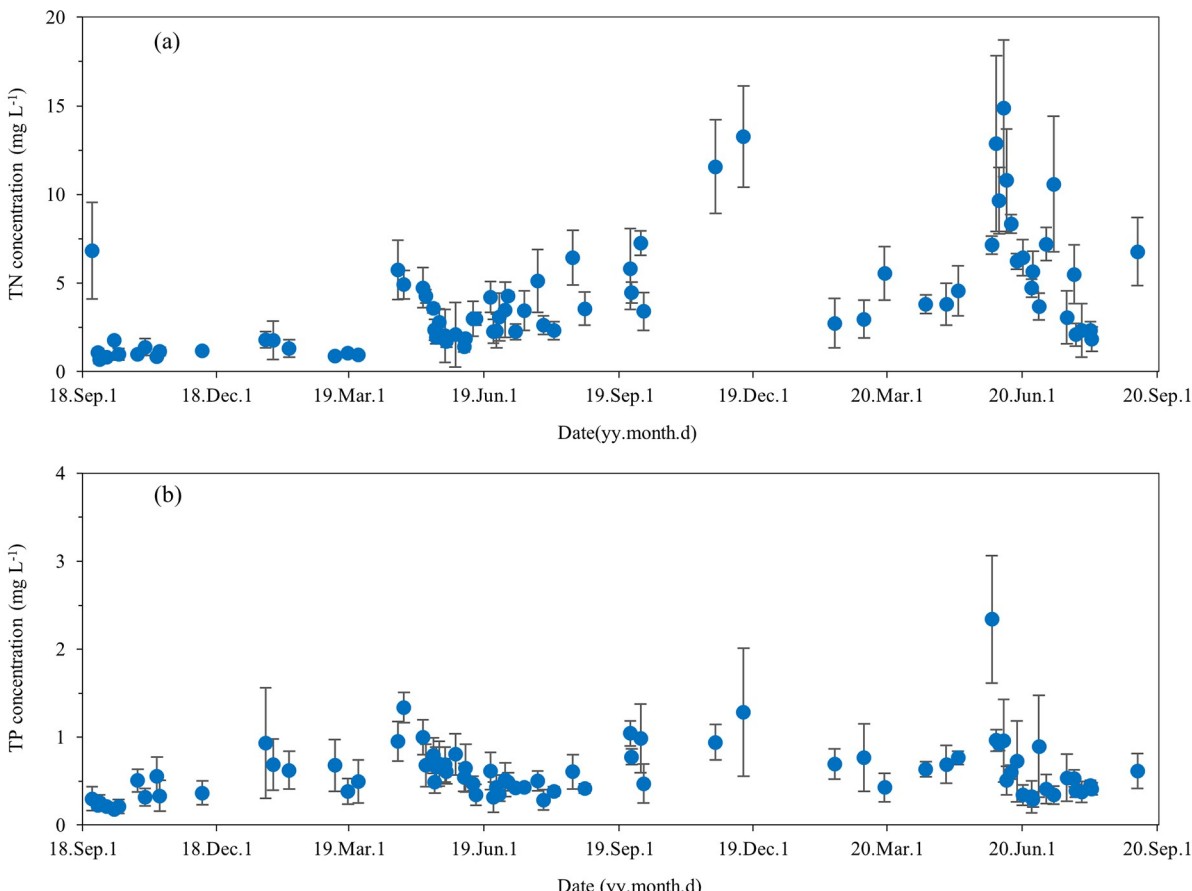

**Fig 3. Daily variations of (a) total nitrogen and (b) total phosphorus concentrations in surface runoff from September 2018 to August 2020.**

The TN loss showed significant differences between the two years (Table 2). The TN losses were 1.08 kg N ha$^{-1}$ in 2018–2019 and 1.85 kg N ha$^{-1}$ in 2019–2020, with an annual average of 1.47 kg N ha$^{-1}$. The runoff losses of TP were both 0.21 kg P ha$^{-1}$ in 2018–2019 and 2019–2020. Therefore, the between-year difference in the TP loss was not significant. During the observation period, the loss rate of fertilizer nitrogen and phosphorus were both 0.43%. The loss rate of nitrogen fertilizer showed significant differences between the two years, with nitrogen loss rate of 0.32% in 2018–2019 and 0.54% in 2019–2020, respectively.

**Table 1. Effects of different rainfall intensities on nitrogen and phosphorus concentrations in surface runoff.**

| Rainfall intensities | N concentration (mg L$^{-1}$) | | | DN/TN (%) | P concentration (mg L$^{-1}$) | | | DP/TP (%) |
|---|---|---|---|---|---|---|---|---|
| | TN | PN | DN | | TP | PP | DP | |
| Light rain | 5.81a | 1.19 a | 4.62 ab | 82.64 a | 0.73 ab | 0.24 a | 0.49 a | 62.23 a |
| Moderate rain | 4.01b | 0.57 c | 3.43 c | 85.23 a | 0.60 bc | 0.22 a | 0.38 ab | 64.21 a |
| Heavy rain | 2.28 c | 0.31 bc | 1.97 b | 86.35 a | 0.44 c | 0.17 a | 0.27 b | 65.63 a |
| Rainstorm | 6.74 a | 0.92 ab | 5.82 a | 82.09 a | 0.79 a | 0.27 a | 0.53 a | 70.21 a |

Note: TN: total nitrogen; PN: particulate nitrogen; DN: dissolved nitrogen; TP: total phosphorus; PP: particulate phosphorus; DP: dissolved phosphorus. Different letters indicate significant differences under different rainfall intensities at $p<0.05$.

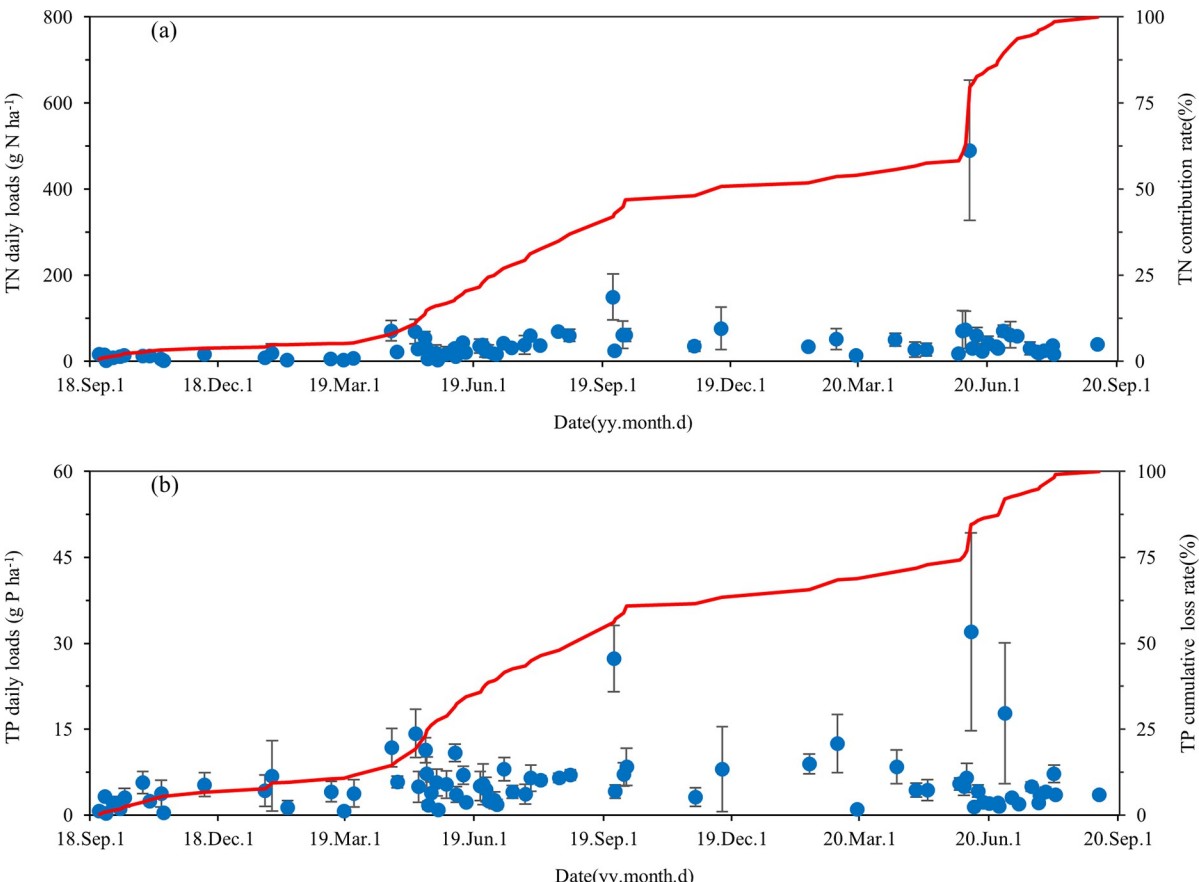

**Fig 4.** The daily loads of (a) TN and (b) TP of each surface runoff event and their contributions to the cumulative losses in the tea plantations from September 2018 to August 2020.

During the 2-year observation period, the TN losses were 0.37, 1.23, 0.57 and 0.77 kg N ha$^{-1}$ under light rain, moderate rain, heavy rain, and rainstorm events, accounted for 12.6%, 41.8%, 19.4%, and 26.2% of cumulative TN losses, respectively (Table 3). Similarly, the TP losses were 0.04, 0.19, 0.11 and 0.08 kg P ha$^{-1}$ under light rain, moderate rain, heavy rain, and rainstorm events, and their contributions to cumulative TP loss were 9.5%, 44.5%, 26.1%, and 19.9%, respectively (Table 3).

## Effect of rainfall intensity on the runoff and nitrogen and phosphorus losses

The pathway analysis described the relationships between surface runoff, total nitrogen and total phosphorus daily load and rainfall intensity (Fig 5A). The results show that rainfall

**Table 2. Cumulative observed TN and TP losses and the loss rates of nitrogen and phosphorus fertilizer.**

| Years | TN loss (kg N ha$^{-1}$ yr$^{-1}$) | Nitrogen fertilizer loss rate (%) | TP loss (kg P ha$^{-1}$ yr$^{-1}$) | Phosphorus fertilizer loss rate (%) |
|---|---|---|---|---|
| 2018–2019 | 1.08 ± 0.14 b | 0.32 ± 0.04 b | 0.21 ± 0.03 a | 0.43 ± 0.05 a |
| 2019–2020 | 1.85 ± 0.31 a | 0.54 ± 0.09 a | 0.21 ± 0.02 a | 0.43 ± 0.04 a |
| 2018–2020 | 1.47 ± 0.10 | 0.43 ± 0.03 | 0.21 ± 0.00 | 0.43 ± 0.00 |

Note: N = 3, different letters indicate significant differences under different monitoring years.

**Table 3. The impact of total nitrogen (TN) and total phosphorus (TP) losses under different rainfall intensities in the tea plantations from September 2018 to August 2020.**

| Rainfall intensities | TN loss | | TP loss | |
|---|---|---|---|---|
| | Amount (kg N ha$^{-1}$) | Contribution rate (%) | Amount (kg P ha$^{-1}$) | Contribution rate (%) |
| Light rain | 0.37 ± 0.04 c | 12.6 | 0.04 ± 0.01c | 9.5 |
| Moderate rain | 1.23 ± 0.08 a | 41.8 | 0.19 ± 0.03 a | 44.5 |
| Heavy rain | 0.57 ± 0.06 bc | 19.4 | 0.11 ± 0.01 b | 26.1 |
| Rainstorm | 0.77 ± 0.21 b | 26.2 | 0.08 ± 0.02 b | 19.9 |

intensity and TN and TP concentrations had a direct positive impact on the amount of TN and TP daily load in surface runoff, which explained 70% of the variation in TN daily load and 78% in TP daily load. The rainfall intensity had a direct positive impact on surface runoff (Fig 5A and 5B, $p < 0.01$), but it did not directly or indirectly affect the TN and TP concentrations in surface runoff. TN daily load was significantly directly affected by surface runoff and TN concentration, and their action coefficients on TN loss were 0.61 and 0.58 ($p<0.01$), respectively. The direct effect of surface runoff on TP loss was stronger than that of TP concentration, and their action coefficients were 0.77 and 0.49, respectively. In this study, the surface runoff and precipitation showed a logarithmic function curve, and the $R^2$ was 0.6597 (Fig 5B, $p < 0.01$).

## Discussion

Tea plants are mainly distributed in tropical and subtropical mountainous areas, where the dominant factors driving annual runoff are high-intensity and high-frequency precipitation [18]. Rainfall intensity magnitude will directly affect the amount of runoff [19]. The study results showed that under light rainfall intensity (10 mm d$^{-1}$) on sloping tea plantation, there is little runoff on the sloping surface, and water flows is intercepted by tea canopy, and from the soil gap down through the cracks underground, which is in line with [20]. Rainfall events above 10 mm contributed significantly to the occurrence of surface runoff in the tea plantation. When the daily rainfall was less than 10 mm, the probability of surface runoff was only 6.8%. When the daily rainfall was greater than 10 mm, the runoff probability of the tea plantations increased to 100%. The rainfall events above 10 mm accounted for only 20.5% of the total

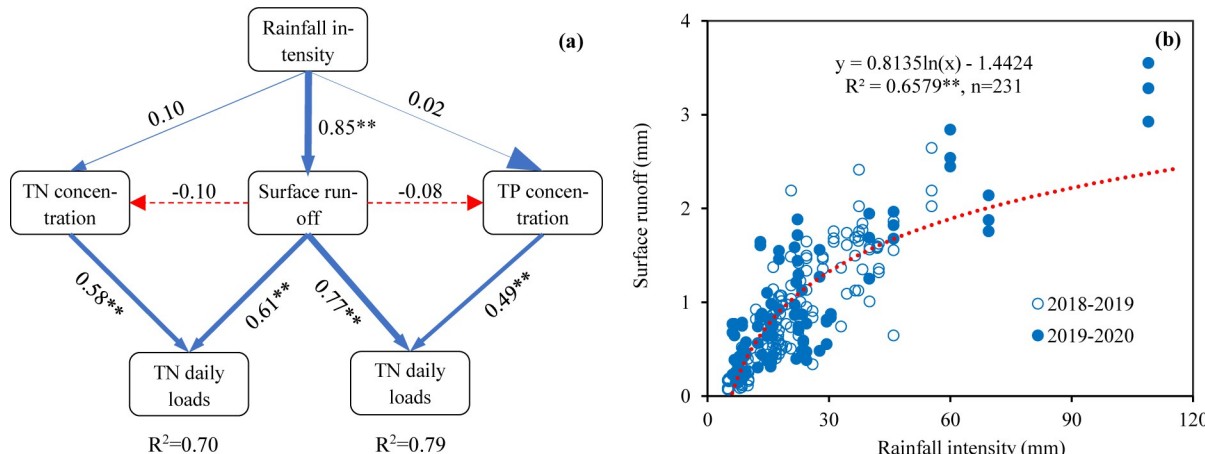

**Fig 5.** (a) Pathway analysis of the rainfall intensity effects on the TN and TP daily loads and (b) the relationship between rainfall intensity and surface runoff. * and ** denoted significant differences in 0.05 and 0.01 levels, respectively.

**Table 4. Nitrogen and phosphorus losses from surface runoff of tea plantations and farmland.**

| Crops | Region | Slope (°) | Rainfall (mm) | Nitrogen input (kg N ha$^{-1}$) | Phosphorus input (kg P ha$^{-1}$) | TN loss (kg N ha$^{-1}$) | TP loss (kg P ha$^{-1}$) | Reference |
|---|---|---|---|---|---|---|---|---|
| Tea | SW China | 12.5 | 1129 | 342.5 | 49.1 | 1.47 | 0.21 | This study |
| Tea | E China | 23.0 | 1380 | 450.0 | 65.4 | 2.19 | 0.23 | [28] |
| Tea | E China | / | 1555 | 454.0 | 51.0 | 2.13 | 0.26 | [26] |
| Tea | C China | 10.0 | 734 | 450.0 | 62.8 | 3.80 | 0.94 | [27] |
| Vegetable | China | / | / | 264.3 | 101.0 | 16.50 | 3.45 | [35] |
| Upland crops | China | / | / | 210.2 | 54.6 | 10.80 | 1.05 | [35] |

rainfall events but 72.6% of the cumulative rainfall. Moreover, 92.5% of the total surface runoff amount occurred in rainfall events above 10 mm. Due to the dispersion of the soil, rainfall enters the soil and infiltrates into it. With the increase of rainfall intensity, soil reaches saturation faster and surface runoff occurs earlier [21]. The surface runoff amount showed synchronization with variations in rainfall erosivity to some extent [22]. With an increase in rainfall intensity, rainfall erosivity is enhanced [23]. Many studies have reported that extreme precipitation events have occurred more frequently and with increasing intensity in recent decades [24, 25]. We can infer that with the further increase in extreme precipitation events in the future, the frequency and amount of surface runoff in tea plantations will increase.

Nutrients can dissolve in surface runoff water and enter surrounding water systems as water flows, causing water pollution and harming human production and life. Therefore, studying the characteristics of water flow on sloping farmland reveals the law of nutrient loss on sloping farmland and provides theoretical support for preventing and controlling the risks caused by nutrient enrichment. The study showed that the total nitrogen and total phosphorus losses in typical tea-producing areas in southwest China were 1.47 kg N ha$^{-1}$ and 0.21 kg P ha$^{-1}$, respectively. The results in this study show similarities with those in the tea plantations in eastern and central China (Table 4) [26–28]. However, there were differences in the characteristics of nitrogen and phosphorus loss in different tea producing regions in China due to the difference of nitrogen and phosphorus nutrient inputs, the terrain conditions and climate characteristics. The surface runoff process of tea plantations is significantly different from that of other crops. Compared with those of grain crops and vegetable systems, the TN and TP loss per unit area of tea plantations were significantly lower [29]. This may be attributed to reduced tillage, the high density of tea trees, and substantial plant litter on the soil surface in tea plantations [20, 30, 31]. These factors are conducive to retaining water and effectively reducing rainwater erosivity, which reduces the occurrence of surface runoff. However, the conversion of forest hillslopes into tea fields may increase nitrogen and phosphorus losses through the surface runoff, particularly during the first years of conversion [32]. Therefore, it is important to be mindful of the potential risks associated with the recent expansion of the tea industry in subtropical China. As a result of this boom, many new tea gardens have been established on hillslopes after the clearance of natural and secondary forests. This trend may have negative consequences for water quality in these areas, which should be carefully monitored and addressed. In addition, it reported that the nutrient loss from the subsurface flow process is much greater than the surface runoff process in hilly areas [33, 34]. Therefore, we can infer that the surface runoff nitrogen and phosphorus losses contribute little to agricultural non-point source pollution in surrounding important water bodies, and more attention should be paid to the subsurface and leaching losses on hills and slopes.

The loss of nitrogen and phosphorus in tea plantations was directly affected by surface runoff and nitrogen and phosphorus concentration in runoff, while rainfall intensity affected the loss of nitrogen and phosphorus by directly driving the occurrence process of surface runoff. The results were similar to other crop systems in the region [36, 37]. Rainfall intensity has a nonlinear effect on the concentration of nitrogen and phosphorus in runoff. Under light rain, moderate rain and heavy rain events, the concentration of nitrogen and phosphorus in runoff continues to decrease with the increase of rainfall intensity. While under rainstorm events, the nitrogen and phosphorus concentrations in runoff were significantly higher than those of moderate rain, and heavy rain events. The percentages of dissolved nitrogen in the total nitrogen lower in the rainstorm than in the heavy rain, which was associated with increased soil erodibility due to the rainfall intensities increase [32]. Our study revealed that winter experiences lower frequency and intensity of precipitation, resulting in lower nitrogen and phosphorus losses after fertilization. Conversely, in spring, precipitation increases, leading to a rapid increase in nitrogen and phosphorus losses after fertilization. The surface runoff occurring during the early stage after fertilizer applications contributed mostly to the total nutrient loss [38]. Applying fertilizers during periods of low precipitation and incorporating them into the soil can reduce the risk of nutrient loss caused by surface runoff [39]. The assessment of the Yangtze River Basin in China shows that the amount of nitrogen loss in drylands is directly affected by rainfall, fertilization, and soil nitrogen content [40]. The interaction between vegetation cover and rainfall characteristics on runoff and nutrient concentrations under different crop systems is extremely complex [41, 42]. Therefore, comprehensively evaluating climate, soil, and nutrient management is necessary to accurately understand the integrity of farmland nitrogen and phosphorus losses on the regional scale.

## Conclusions

Based on our two-year observation of runoff plot data, this study quantitatively evaluated the characteristics of surface runoff and nitrogen and phosphorus losses of tea plantations in the mountainous area of southwest China. The results showed that the TN and TP losses in the tea plantations were 1.47 kg N ha$^{-1}$ yr$^{-1}$ and 0.21 kg P ha$^{-1}$ yr$^{-1}$, and the loss rate of nitrogen and phosphorus fertilizer were both 0.43%, respectively. This experimental study illustrated that surface runoff will not cause a large amount of nitrogen and phosphorus loss from tea plantations in the mountainous areas of southwest China. Rainfall intensity had a direct and significant impact on surface runoff and TN and TP nutrient loss. In the rainfall events above 10 mm, the total surface runoff accounted for 92.5%, the TN loss accounted for 87.4%, and the TP loss accounted for 90.5%. Especially under rainstorm events, the surface runoff was large, the nitrogen and phosphorus concentrations in the runoff were high, and the accumulation of nitrogen and phosphorus losses was large. The path analysis also showed that runoff and nitrogen and phosphorus concentrations directly and actively promoted the nitrogen and phosphorus losses. These had the most significant impact on water pollution and nutrient loss in tea plantations. In the tea production process in southwest China, proper control of rainfall erosion is key to preventing and controlling surface runoff and nutrient loss. By controlling the generation of surface runoff and avoiding excessive fertilization during the rainy season, nutrient loss can be controlled. However, the processes and underlying mechanisms of N and P migration to surface and groundwater are still unclear and require further study on a large scale. To achieve high-yield, high-quality tea while minimizing nitrogen and phosphorus pollution in tea production, optimizing nutritional management is of paramount importance in the mountainous area of southwest China.

## Supporting information

**S1 Table. Characteristics of annual rainfall events and erosion runoff events during the observation period.**
(DOCX)

**S2 Table. The difference of runoff events and erosion runoff events under different rainfall intensities in the tea plantations from September 2018 to August 2020.**
(DOCX)

**S3 Table. Characteristics of runoff events, runoff amounts and runoff coefficient during the observation period.**
(DOCX)

**S4 Table. The difference of runoff events, runoff amounts and runoff coefficient under different rainfall intensities in the tea plantations from September 2018 to August 2020.**
(DOCX)

## Acknowledgments

The authors are grateful to the editor and two anonymous reviewers for their advice in improving this paper.

## Author Contributions

**Conceptualization:** Xingcheng Huang, Yu Li, Taiming Jiang.

**Data curation:** Xingcheng Huang.

**Formal analysis:** Xingcheng Huang.

**Funding acquisition:** Taiming Jiang.

**Investigation:** Darong Zhen, Xiaona Lu.

**Methodology:** Yarong Zhang, Yanling Liu, Yu Li.

**Project administration:** Darong Zhen, Yu Li.

**Resources:** Xiaona Lu.

**Software:** Xingcheng Huang.

**Supervision:** Darong Zhen, Yu Li.

**Validation:** Yarong Zhang, Yanling Liu.

**Visualization:** Xingcheng Huang.

**Writing – original draft:** Xingcheng Huang.

**Writing – review & editing:** Xingcheng Huang, Yu Li, Taiming Jiang.

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
