## [Decision Letter · Decision Letter 0]

28 Nov 2022

PONE-D-22-23175Nitrogen and Phosphorus Losses via Surface Runoff from Tea Plantations in the Mountainous Areas of Southwest ChinaPLOS ONE

Dear Dr. Huang,

Thank you for submitting your manuscript to PLOS ONE. After careful consideration, we feel that it has merit but does not fully meet PLOS ONE’s publication criteria as it currently stands. Therefore, we invite you to submit a revised version of the manuscript that addresses the points raised during the review process.

The manuscript was reviewed by two reviewers, both of who found major problems with the manuscript and the analysis performed in the study. If the authors chose to revised and resubmit, the manuscript should be completely re-written and re-organised. The statistical analysis should be reworked and completely described. The results of the study need to highlight the innovative nature of the study, with respect to similar studies in other regions. To be published, the manuscript needs to be clearly useful to journal readers. At the moment, while there appears to be potential, the present manuscript does not show sufficient innovation. Furthermoer, the manuscript has problems of organisation and English, both take away from the clarity of the text.

We look forward to receiving your revised manuscript.

Kind regards,

Steven Arthur Loiselle

Academic Editor

PLOS ONE

Journal Requirements:

"This research was funded by the National Natural Science Foundation of China (31860132), the Program of Science and Technology Plan of Guizhou Province (20182340 and 20201Y119) and the Program of Science and Technology Innova-tion Talents Team of Guizhou Province (20185604)."

"This research was funded by the National Natural Science Foundation of China (31860132), the Program of Science and Technology Plan of Guizhou Province (20182340 and 20201Y119) and the Program of Science and Technology Innovation Talents Team of Guizhou Province (20185604).

4. We note that you have stated that you will provide repository information for your data at acceptance. Should your manuscript be accepted for publication, we will hold it until you provide the relevant accession numbers or DOIs necessary to access your data. If you wish to make changes to your Data Availability statement, please describe these changes in your cover letter and we will update your Data Availability statement to reflect the information you provide

5. We note that Figure 1 in your submission contain copyrighted images. All PLOS content is published under the Creative Commons Attribution License (CC BY 4.0), which means that the manuscript, images, and Supporting Information files will be freely available online, and any third party is permitted to access, download, copy, distribute, and use these materials in any way, even commercially, with proper attribution. For more information, see our copyright guidelines: http://journals.plos.org/plosone/s/licenses-and-copyright.

Additional Editor Comments (if provided):

The manuscript was reviewed by tow reviewers, both of who found major problems with the manuscript and the analysis performed in the study. If the authors chose to revised and resubmit, the manuscript should be completely re-written and re-organised. The statistical analysis should be reworked and completely described. The results of the study need to be innovative and useful to the journal readers. At the moment, while there appears to be potential, the present manuscript does not highlight sufficient innovation. Furthermoer, the manuscript has problems of organisation and English, both take away from the clarity of the text.

Reviewers' comments:

Reviewer's Responses to Questions

**Comments to the Author**

1. Is the manuscript technically sound, and do the data support the conclusions?

Reviewer #1: Partly

Reviewer #2: Yes

2. Has the statistical analysis been performed appropriately and rigorously? 

Reviewer #1: No

Reviewer #2: Yes

3. Have the authors made all data underlying the findings in their manuscript fully available?

Reviewer #1: No

Reviewer #2: Yes

4. Is the manuscript presented in an intelligible fashion and written in standard English?

Reviewer #1: Yes

Reviewer #2: Yes

5. Review Comments to the Author

Reviewer #1: This is an interesting study on nitrogen and phosphorus losses from tea plantations. However, I found several issues related to data processing and interpretation, statistics and writing style.

My major concerns:

1) The authors did not comprehensively explain their statistical approach in the methods section. There is no information on the statistical models, data transformation, mean separation and "pathway analysis." I was lost when they brought the results from the abovementioned analysis in the results and discussion sections. They have also used statistical terms like "significant" without presenting a test or context. Many comparisons were also made without a statistical approach. I urge authors to take a second look at their manuscript, update statistical methods and tests and base the interpretations on the statistical tests.

2) Crucial information is missing in sample collection and data processing. The authors did not mention whether the presented concentration is daily or flow-weighted.

3) Figure 3/4. Adding daily loads as a panel under one of these figures would be beneficial, and it will also help to see if large events are associated with major nutrient losses.

4) Avoid repeating results in the discussion section.

5) Avoid repeating your results in the conclusion section.

Line 34: Bouwman et al. 2002

Line 35: Phosphorus accumulation

Line 37: Please replace "chiefly" with "primarily"

Line 38: "emissions" generally indicate "gaseous discharge." Here you are mainly discussing "water discharge." Please use more appropriate terminology.

Line 55-58: Briefly discuss the findings from previous studies that assessed runoff pollution from tea plantations.

Line 76: What is the spacing between the plots?

Line 98: No "statistical tests" are mentioned here. Briefly explain data curation, distribution and transformation, statistical analysis, mean separation, and p-value for statistical significance.

Line 111: Please delete "much"

Line 115: In the methods section, please explain how you differentiated "erosive rainfall events" from "non-erosive rainfall" events.

Line 123: Please explain how you define surface runoff events in the methods section. As per your figure 3, there were periods where runoff occurred for two or more consecutive dates. Were they considered single events?

Line 133: Did you use "daily flow-weighted mean concentration" or "daily concentration"? Unclear.

Line 135: Can you support this claim with a statistical test?

Line 139: Ensure the spacing between the values and units is consistent.

Figure 4: Seasonal variation or daily variation? What do error bars represent?

Table 1. Section 2.4 statistical analysis did not provide any information about the tests you performed and mean separation methods. We can evaluate your results and interpretation without this critical information.

Also, does "daily mean concentration"?

Line 145: Provide the statistical model and p-value to support this claim.

Line 147: Wrong. Your N and P concentration were similar for light rain and rainstorm based on your statistical analysis (which we have no idea as it is not provided). Also no difference in PP concentration among all rainfall events.

Line 163: You bring this "pathway analysis" from nowhere. There is nothing written about this in the materials and methods. So many terms (e.g., action coefficient) and symbols (e.g., lambda) were being forced here without any previous explanations, which could not be very clear to readers.

Figure 5: What precipitation, event, daily or cumulative? Be specific.

Line 191: Avoid mentioning results in the discussion. Also, you cant throw these numbers without a range and context. Numbers change depending on weather, topography and management.

Line 194: Delete. You are repeating the same information in the next sentence.

Line 195: You often use "significant" without a test or supporting evidence.

Table 5: You can add the "slope/topography" of the plantations to this table, which may explain some of the observed variability.

Line 199- it was reported

Line 202- pollution from tea plantations in Southwest China

Line 210- Not entirely true. Overland flow is generated when the intensity exceeds the soil's infiltration capacity. It doesn't necessarily mean that the soils are saturated at that time. Revise.

Lines 211-213 – Merge these sentences

Line 220- Cropping systems

Line 236- delete the sentence "according to…."

Reviewer #2: The authors investigated the Nitrogen and Phosphorus Losses via Surface Runoff from Tea Plantations in the Mountainous Areas of Southwest China. Their results are potentially interesting and useful. I may have the following major and minor comments.

1. The key results from your abstract is not clear. You need to be very sharp on your key conclusions.

2. Line 26-29. The conclusions and implications are not much attractive. They should be built on your key results.

3. It is unclear from your introduction that why do you want to investigate Nitrogen and Phosphorus Losses? Why are they important? Why do you choose tea plant cultivation? See relevant publication, Sheteiwy et al., 2022, https://doi.org/ 10.1016/j.envpol.2022.120356; Zhou et al., https://doi.org/10.1016/j.scitotenv.2019.133845; Xu et al., https://doi.org/10.1016/j.spc.2021.04.019

4. There is quite large room to improve the writing. Not only for the writing itself but also for the writing logic. The research questions and hypotheses are not clear.

5. To make your results comparable, more details on Materials and methods are required for the further evaluation. More information on the climatic, edaphic and environmental variables are required.

6. More information on the land use history and plant and understory community composition are required. More details on the cover crop management are required.

7. Relevant citations are required for your method section.

8. Several unnecessary abbreviations are preventing the reading. I have to remember a lot abbreviations when reading your manuscript.

9. You need more efforts for the results section to well focus on your key findings. This can make your results clearer.

10. Some in-depth data analyses and particularly data interpretation is required to explore the underlying mechanisms. Will you compare the relative importance of these variables? Have you tried to discuss the causal relationship?

11. You need to clearly mention the hypotheses behind your SEM.

12. Tables. Decimals should be consistent across the manuscript.

13. Some sentences are unnecessary long with changing focuses. It is not easy to understand these long sentences. The writing needs to be improved.

14. The main conclusions and key implications are not clear enough from your discussion. The potential mechanisms need to be discussed. What are the important biotic and abiotic factors affecting the responses? What are the key implications? How can we advance the understanding of N and P cycling from this study? For example, Luo et al., 2022, https://doi.org/10.1111/1365-2435.14178; Ren et al., 2017, https://doi.org/10.1007/s00374-017-1197-x; Jiang et al., https://doi.org/10.1016/j.fcr.2019.02.010

15. Fig. 1. Coordinates are required.

6. PLOS authors have the option to publish the peer review history of their article (what does this mean?). If published, this will include your full peer review and any attached files.

Reviewer #1: No

Reviewer #2: No

---

## [Author Response · Author response to Decision Letter 0]

4 Feb 2023

Dear Sir,

Responses to Reviewers comments on PONE-D-22-23175

We thank the reviewers for their overall very positive and encouraging assessment and their constructive comments, which helped us to significantly improve the manuscript. The changes are updated on the current manuscript. We trust that this version is in good shape for acceptance. Below are the detailed responses for each of the suggested changes.

---

## [Decision Letter · Decision Letter 1]

22 Mar 2023

PONE-D-22-23175R1Nitrogen and phosphorus losses via surface runoff from tea plantations in the mountainous areas of Southwest ChinaPLOS ONE

Dear Dr. Huang,

Thank you for submitting your manuscript to PLOS ONE. After careful consideration, we feel that it has merit but does not fully meet PLOS ONE’s publication criteria as it currently stands. Therefore, we invite you to submit a revised version of the manuscript that addresses the points raised during the review process.

We look forward to receiving your revised manuscript.

Kind regards,

Steven Arthur Loiselle

Academic Editor

PLOS ONE

Journal Requirements:

Additional Editor Comments:

The manuscript has been significantly improved, however there are still multiple areas that are not clear. Importantly, the Conclusions remain a summary of the results already presented early in the manuscript, rather than conclusions that address the implications of the study for planning or managment (eg.with respect to climate changes). the conclusions should also better address challenges and future studies. A reviewer has also listed a number or improvements that should be made.

Reviewers' comments:

Reviewer's Responses to Questions

**Comments to the Author**

1. If the authors have adequately addressed your comments raised in a previous round of review and you feel that this manuscript is now acceptable for publication, you may indicate that here to bypass the “Comments to the Author” section, enter your conflict of interest statement in the “Confidential to Editor” section, and submit your "Accept" recommendation.

Reviewer #1: All comments have been addressed

2. Is the manuscript technically sound, and do the data support the conclusions?

Reviewer #1: Yes

3. Has the statistical analysis been performed appropriately and rigorously? 

Reviewer #1: Yes

4. Have the authors made all data underlying the findings in their manuscript fully available?

Reviewer #1: No

5. Is the manuscript presented in an intelligible fashion and written in standard English?

Reviewer #1: Yes

6. Review Comments to the Author

Reviewer #1: I thank the authors for their effort and time to revise the manuscript. Some minor questions:

Line 19- please delete “the results show that”

Line 71: when the fields were fertilized? (Follow up in discussion: Did high concentration coincide with fertilizer application dates? Some studies have shown high nutrient conc. in runoff occurred immediately following fertilization. A comment or two would be nice).

Line 79 and elsewhere – space between the word and (

Line 87 – how long the samples were stored in -4oC before being analyzed?

Line 88- how did you wash the tanks.

Line 102- mention your fixed, random and repeated factors in your analysis

Line 142 – spacing was not consistent between the values and units

Table 1. Describe the abbreviations in your title

Line 158 – delete “clearly”

Line 206- did you notice that in your study? You have measured dissolve and total fractions. This could give some insights.

Line 220- reduced tillage/conservation tillage?

7. PLOS authors have the option to publish the peer review history of their article (what does this mean?). If published, this will include your full peer review and any attached files.

Reviewer #1: No

---

## [Author Response · Author response to Decision Letter 1]

24 Mar 2023

Dear reviewer,

Thank you very much for your helpful comments on our manuscript. The changes are updated on the current manuscript. We trust that this version is in good shape for acceptance. Below are the detailed responses for each of the suggested changes.

1.Line 19,Noted and deleted.

2.Line 71,We thank the reviewer for this comment, and following the reviewer’s suggestion. We have discussed fertilizer application to concentration of nitrogen and phosphorus in surface runoff.

3.Line 79,Noted and correction effected.

4.Line 87,All runoff samples were generally analyzed within one week after collection.

5.Line 88,This is well noted and supplemented in Materials and Methods to improve clarity.

6.Line 102,ANOVAs were performed with fertilizer application as the fixed factor, and rainfall intensities as the random factor, and three runoff plots and two observation years as repeated factors.

7.Line 142,Noted and correction effected.

8.Table 1,Noted and correction effected.

9.Line 158,Noted and deleted.

10.Line 206,We thank Reviewer for critical suggestions. We discussed the differences in the dissolved fractions of nitrogen and phosphorus in runoff under different rainfall intensities.

---

## [Editor Report · Decision Letter 2]

16 Apr 2023

Nitrogen and phosphorus losses via surface runoff from tea plantations in the mountainous areas of Southwest China

PONE-D-22-23175R2

Dear Dr. Huang,

We’re pleased to inform you that your manuscript has been judged scientifically suitable for publication and will be formally accepted for publication once it meets all outstanding technical requirements.

Kind regards,

Steven Arthur Loiselle

Academic Editor

PLOS ONE
---

## [Editor Report · Acceptance letter]

19 Apr 2023

PONE-D-22-23175R2 

Nitrogen and phosphorus losses via surface runoff from tea plantations in the mountainous areas of Southwest China 

Dear Dr. Huang:

I'm pleased to inform you that your manuscript has been deemed suitable for publication in PLOS ONE. Congratulations! Your manuscript is now with our production department. 

Kind regards, 

on behalf of

Dr. Steven Arthur Loiselle 

Academic Editor

PLOS ONE